# Early Rate of Force Development and Maximal Strength at Different Positions of the Athletic Shoulder Test in Baseball Players

**DOI:** 10.3390/sports13090300

**Published:** 2025-09-01

**Authors:** Ben Ashworth, Mikulas Hank, Omid Khaiyat, Ginny Coyles, Ferdia Fallon Verbruggen, Erika Zemkova, Frantisek Zahalka, Tomas Maly

**Affiliations:** 1School of Health and Sport Sciences, Liverpool Hope University, Liverpool L16 9JD, UK; ben@athleticshoulder.com (B.A.); alizado@hope.ac.uk (O.K.); coylesg@hope.ac.uk (G.C.); 2Sport Research Centre, Charles University, 162 52 Prague, Czech Republic; ferdiafv@gmail.com (F.F.V.); zahalka@ftvs.cuni.cz (F.Z.); tomimaly@yahoo.com (T.M.); 3Department of Biological and Medical Sciences, Faculty of Physical Education and Sports, Comenius University in Bratislava, 814 69 Bratislava, Slovakia; erika.zemkova@uniba.sk

**Keywords:** throwing, ASH test, shoulder, athletes, isometrics, performance, injury prevention

## Abstract

Background/Objectives: Peak force (PF) reflects maximal strength, while early rate of force development (RFD; 0–100 ms) indicates explosive neuromuscular output. The Athletic Shoulder (ASH) test is gaining popularity in overhead athlete profiling, but its use for assessing explosive strength in various shoulder positions is underexplored. This study compared PF and RFD at shoulder abductions of 180° (ASH-I), 135° (ASH-Y), and 90° (ASH-T) in baseball players. Methods: Seventeen male athletes (age 22.7 ± 4.2 years; height 186.3 ± 7.3 cm; body mass 83.9 ± 10.1 kg) performed isometric ASH tests with the dominant arm. PF, PF relative to body mass (PF/BM), and early RFD were analysed. Results: ASH I showed 25% significantly higher PF (182 ± 41 N), PF/BM (2.15 ± 0.39 N/kg), and 40% higher RFD (545 N/s) than ASH Y or T (all *p* < 0.001), which did not differ significantly. PF showed excellent reliability (ICC = 0.86–0.93); RFD showed moderate-to-good reliability (ICC = 0.75–0.81). Smallest worthwhile changes were ~5% for PF and ~15% for RFD. Conclusions: Maximal isometric shoulder strength and explosiveness were highest at 180° abduction in baseball athletes, with no significant difference between 135° and 90°. PF demonstrated excellent reliability, while early RFD showed moderate to good reliability and higher variability, highlighting the need for repeated measures. These findings provide specific position reference values and support the inclusion of multiple abduction angles in shoulder strength assessment to detect neuromuscular deficits and monitor training adaptations in baseball athletes.

## 1. Introduction

High power production in the shoulder is an important indicator of elite throwing performance in overhead sports like baseball, while development of upper body maximal strength and ability to produce high force as fast as possible result in factors like fatigue resistance (increasing tolerance to training/competition load), ball velocity (over 140 km/h), and injury prevention [1,2,3,4]. Optimization of the kinetic chain is dependent on neuromuscular coordination and force transfer across lower limbs, pelvis, trunk rotation, and upper limbs, resulting in shoulder rotation angular velocities of over 6000 °/s [5,6,7,8]. As throwing is a complex high-intensity motion—mainly in the arm acceleration phase, which happens in milliseconds—rate of force development (RFD), and its early phase (measured from the beginning of contraction up to 100 ms), is becoming recognized as a more sensitive parameter of power potential when compared to traditional peak force (PF), or even the late-phase RFD (from 100 up to 250 ms) evaluated by clinicians [9,10,11,12]. However, examples of RFD shoulder performance data are significantly lower in sports science in comparison to the lower limbs, despite higher angular velocities [13]. The lack of performance monitoring in overhead sports like baseball, where injuries to the shoulder and elbow are costly, can not only negatively influence performance but also impact athlete’s careers [14,15]. When an injury occurs, it often takes a long recovery time, and athletes return to competition in sub-optimal condition [15,16].

Field-based isometric strength assessments like the Athletic Shoulder (ASH) test are the most common methods for relatively quick and safe athlete monitoring on a weekly basis [17]. The ASH test demonstrated high-to-excellent validity and reliability as a tool for PF evaluation in three arm positions (ASH I in 180° shoulder abduction; ASH Y in 135° shoulder abduction; ASH T in 90° shoulder abduction). While recent research has started to also examine RFD parameters in overhead athletes [18,19], it is important to note that, based on higher variability of early RFD results in studies, this parameter seems to be more sensitive to technical execution, individual neuromuscular coordination, and test familiarization when compared to PF [20]. Because of this, consistent test–retest conditions, body segment positions, familiarization, and verbal explanation of producing force as fast as possible is important when monitoring athletes more often and evaluating early RFD. Volleyball athletes have been shown to have close power performance relationships between ASH I early RFD and ball velocity during the serve [18], but there is a lack of data or supportive evidence in sports like baseball, especially regarding normalization of forces to athlete’s body mass, or comparisons of RFD differences between the three ASH test positions.

In baseball pitching, shoulder abduction angles typically range from approximately 90° at ball release increasing during maximal shoulder external rotation in the late cocking phase [5,6]. At higher angles (>90°), neuromuscular coordination between deltoid muscles and rotator cuff muscles is important for maintaining glenohumeral stability and effective force transfer [21]. Each of the ASH test positions corresponds to a portion of this range, as ASH I (180° of shoulder abduction), ASH Y (135° of shoulder abduction) reflects the transition to acceleration, and ASH T (90° of shoulder abduction) reflects the arm position at ball release [5,22]. This alignment suggests that the ASH test may offer greater ecological validity for baseball than short-lever internal and external shoulder rotation assessment, as it replicates the long lever and higher shoulder abduction during pitching. While previous research in volleyball players has linked ASH I early RFD to serve velocity [18], baseball differs in that the ball must be released after being carried through an optimal acceleration trajectory, potentially creating distinct neuromuscular demands that justify specific position strength assessment. Comparing three ASH test positions in baseball athletes can reveal specific position optimal results or deficits, inform targeted conditioning, and improve the sensitivity of monitoring programs for performance optimization and injury prevention.

Thus, this study examined PF and early RFD at different ASH test shoulder positions of high-level baseball athletes. The hypothesis is that there are significant differences (*p* < 0.05) in force performance between the three arm positions in the ASH test, particularly with the highest in ASH I position and lowest in the ASH T position. The results should enhance the knowledge about how, and to what degree, different shoulder abduction angles may change performance in baseball athletes. Although ASH test PF reliability was previously established [19], the data about consistency of early RFD is limited and underexplored within the context of ASH test protocols. Given its sensitivity to individual movement strategies in explosive strength performance [20], results expect early RFD to show lower, but acceptable, reliability compared to PF. The data should help with interpretation of ASH test results for coaches, clinicians, or researchers when profiling athletes, monitoring during rehabilitation or high competition loads, or making return-to-performance decisions.

## 2. Materials and Methods

### 2.1. Participants

A total of seventeen baseball athletes (age: 22.7 ± 4.2 years; body height: 186.3 ± 7.3 m; body mass: 83.9 ± 10.1 kg) were analysed in this research. Participants were active athletes of the Czech national baseball team, including 12 pitchers and 5 position players (3 infielders and 2 outfielders). Preferred throwing arm was selected as the dominant arm for testing by verbal questioning. The inclusion criteria were as follows: baseball male athletes; minimum of six years of training in the highest national competition; and minimum of two years of being a of national team member. The exclusion criteria were as follows: any signs of pain or musculoskeletal injuries excluding them from training/competition; high-intensity throwing or strength training within last 48 h; and having not performed the ASH test in the past. Participants were asked to not perform any high-intensity training or throwing activity within 48 h prior to strength assessment and were familiarized with the ASH test by completing a minimum of three tests on different days prior to the data collection. The study was conducted in accordance with the Declaration of Helsinki and approved by the Institutional Ethical Committee of the Liverpool Hope University under n. S11-06-19PA049. Informed consent was obtained from all subjects involved in the study.

### 2.2. Procedures

#### Isometric Shoulder Flexion Strength Assessment

The ASH test was conducted in accordance with the original isometric strength assessment protocol [17]. Participants performed maximal isometric contractions in three arm positions—ASH I at 180° abduction, ASH Y at 135° abduction, and ASH T at 90° abduction—with pronated forearm and heel of the hand as the primary contact points on the force platform (Figure 1). Only the dominant upper limb was tested, defined as the athlete’s preferred throwing arm, which was identified by verbal questioning. Consistent across all three positions, the elbow was kept as straight as possible, and excess scapula elevation was avoided. All ASH tests were performed with the participant lying prone, with their forehead positioned on a 4 cm pad for neck standardization. The non-test arm was placed behind the back during ASH T and ASH Y tests to stop fixation with hand or elbow on the floor. For the ASH I position, the opposite arm was placed by the side on the floor. Hand position on the force platform was consistent without resting the forearm on the plate. A vertical axis force plate system ForceDecks (Vald, Brisbane, Australia) connected to proprietary data acquisition and analysis software was positioned on the floor adjacent to the participant’s shoulder and used to measure force output. After the individual players’ throwing warm up, followed by 2 submaximal (80–90% physical effort) contractions in each test position, participants performed 3 maximal trials in each position on the dominant limb, separated by a 20 s rest. Each maximal voluntary contraction was held for approximately 3 s to allow for assessment of both RFD (0–100 ms) and PF within the same trial. The 0–100 ms window was selected because it reflects the early explosive force production phase, which is most relevant to the rapid arm acceleration in baseball pitching and throwing, typically occurring within 150 ms [20]. The order of testing positions was consistently I, Y, T, as recommended by the original ASH test protocol [17] and commonly applied by clinicians. Participants received standardized instructions [20] and consistent verbal encouragement during each trial, including push as “fast and hard” as possible. A verbal countdown was provided prior to each effort. Trials were excluded and repeated if there was an instance of a failure to perform the test according to the instructions.

### 2.3. Data Acquisition and Processing

Force–time data were collected at a sampling frequency of 1000 Hz by force plate system ForceDecks (Vald, Brisbane, Australia) and exported for analysis. Signals were filtered using a 4^th^-order Butterworth low-pass filter with a 20 Hz cutoff. Contraction onset was defined as the point where force exceeded baseline by more than 5 N within a 0.2 s window. Early RFD was calculated as the change in force from contraction onset to the force value at 100 ms, divided by the corresponding time interval. PF was defined as the highest instantaneous force value recorded within the trial.

### 2.4. Statistical Analysis

Violation of normality assumption was not confirmed by the Shapiro–Wilk test (*p* < 0.05) for all PF data; thus, descriptive statistics are presented as means with standard deviations. Due to violations of normality assumptions confirmed by the Shapiro–Wilk test (*p* < 0.05) for all RFD data, descriptive statistics are presented as medians with interquartile range of 25th and 75th percentile. Repeated-measures analysis of variance (ANOVA) was performed to evaluate within-subject differences in PF and PF/BM (dependent variable) across the three ASH test positions (independent variable). Paired t-tests between all pairs of shoulder positions with Bonferroni correction was used to evaluate multiple comparisons. A non-parametric repeated-measures ANOVA was performed using Friedman test to evaluate within-subject differences in early RFD (dependent variable) across the three ASH test positions (independent variable). Wilcoxon signed-rank post hoc test with Bonferroni correction was used to evaluate pairwise comparisons. A significance level of *p* < 0.05 was used for all statistical tests. Coefficient of variation (CV), standard error of measurement (SEM), and intraclass correlation coefficients (ICCs) were calculated individually for PF and RFD to evaluate the relative reliability of repeated measures in each ASH test shoulder position. ICCs were classified as follows: <0.50 as poor; 0.50–0.75 as moderate; 0.75–0.90 as good; and >0.90 as excellent [19]. For the comparison, only the best repetition values were chosen to reduce variability from submaximal efforts and reflecting the trial maximal force production within explosive neuromuscular task. All statistical analyses were executed using IBM^®^ SPSS^®^ v21 (Statistical Package for Social Science, Inc., Chicago, IL, USA, 2012) and Python (v3.12.3; Python Software Foundation, Wilmington, DE, USA) for data processing, visualization, and inferential testing.

## 3. Results

Descriptive PF, PF/BW, and early RFD data for each of the three shoulder ASH test positions are shown in Table 1. Significant differences were found between ASH test shoulder positions in PF variable (F = 21.15; df = 2; *p* < 0.001) (Figure 2). The ASH I position reached 24% higher (*p* < 0.05) results in PF performance when compared to ASH Y and 25% higher (*p* < 0.05) than ASH T position. Figure 2 shows the smallest worthwhile change (SWC) in PF variable was 4%, while the minimal detectable change (MDC) reached around 15%.

Significant differences between the ASH test positions in PF/BM (F = 20.36; df = 2; *p* < 0.001) are shown in Figure 3. In PF/BM, differences were found between ASH I and ASH Y in about 24% (*p* < 0.001) and 25% (*p* < 0.001) of cases when compared to ASH T. Figure 3 shows the SWC in PF/BM is approximately also about 4%, while for MDC, it ranged from 10 to 14%.

Significant differences were found between the ASH test shoulder positions with the early RFD variable (χ^2^ = 13.94; df = 2; *p* < 0.001) (Figure 4). The ASH I position yielded 44% higher (*p* < 0.05) results in early RFD performance when compared to ASH Y, and the results were 38% higher (*p* < 0.05) than those for the ASH T position. Figure 4 shows the SWC in early RFD ranges from 14 to 19%, while for MDC, it ranged from 43 to 75%.

PF data showed excellent test–retest reliability across all ASH positions (ASH I ICC = 0.93; ASH Y ICC = 0.86; ASH T ICC = 0.93). RFD showed good to moderate reliability (ASH I ICC = 0.77; ASH Y ICC = 0.75; ASH T ICC = 0.81). The coefficient of variation for PF across all ASH test positions ranged from CV = 1.0% to 13.7%, with most values being below 10%, indicating good to excellent within-subject consistency. RFD demonstrated higher variability, with values ranging from CV = 3% to over 160%. Standard error of measurement (SEM) were relatively low for PF (most results below 7 N) but higher for RFD (exceeding 50 N/s).

## 4. Discussion

The aim of this study was to examine the differences in force output between three ASH test shoulder positions in baseball athletes. The results suggest that shoulder position affects the ability of athletes to generate maximal force at maximal rates differently. The main finding was that ASH I position in 180° shoulder abduction was significantly higher in maximal force and early RFD when compared to other shoulder positions in lower abduction angles (*p* < 0.05). Our analysis found no difference in force between ASH Y in 135° abduction and ASH T position in 90, while the ASH T position yielded slightly higher (12%) early RFD values as ASH Y, although the differences were non-significant.

Overhead throwing phases involve high abduction angles (>90°) and explosive movements, requiring maximal force closure from the deltoid and rotator cuff force couple to stabilize the joint and prevent injury [21,22,23]. The ASH test’s higher abduction and long lever arm positions replicate the shoulder joint loading pattern and muscle coordination demands observed in the late cocking and acceleration phases of baseball pitching, where shoulder abduction angles range from approximately 80° to 125° and above [5,22,23]. These shoulder positions need similar contributions from the deltoid and rotator cuff muscles and scapular stabilizers [24,25], supporting the ecological validity of using the ASH test for PF and early RFD measures for performance monitoring, rehabilitation, and return-to-play decision making in baseball. Further research emphasizes joint stabilization during high-velocity throwing, as at higher abduction angles, the need for coordination between deltoid, rotator cuff, and scapular stabilizers is significant to maintain glenohumeral stability and optimize force transfer [24,25]. As found in Hecker et al.’s study [22], the deltoid’s contribution to abduction strength increases linearly with abduction angle, from about 24% at 0° to 75% at 120°, indicating significantly greater involvement at higher angles (as in throwing). The middle deltoid provides the largest compressive force during abduction (a force that pushes the humeral head into the glenoid fossa), but especially at higher angles, it may also generate significant shear force (force that acts parallel to the joint surface, which can potentially destabilize the joint or stress soft tissues, especially if rotator cuff strength is insufficient) [23,25]. The rotator cuff muscles like supraspinatus, infraspinatus, and subscapularis are important for joint stability during abduction in the transverse plane. Their activation counteracts previously mentioned shear force from deltoid, particularly as abduction angle increases [23,25,26]. Muscles like the pectoralis major and latissimus dorsi contribute synergistically with the rotator cuff to humeral head stability, but due to their moment, the arms also act as scapular-plane destabilizers that create glenohumeral joint superior/inferior shear force, especially as rotator cuff demand increases with higher abduction angles [26,27,28]. The longer lever arm in the ASH test may more closely replicate a throwing specific action than short lever tests of rotation (ER/IR). The increasing shoulder joint torques in the high-threshold ASH test mean that shear forces from the deltoid, latissimus dorsi, and pectoralis major demand a higher level of force production from the rotator cuff, making the ASH test ideal to test maximal force capability across the shoulder girdle.

For time-constrained explosive movement actions like throwing, where arm acceleration happens in under 150 ms, early RFD (within the first 100–150 ms) seems more responsive to training and performance changes than PF [29]. For example, improvements in early RFD were linked to better jumping performance, while PF did not show significant changes [29]. Other research has also confirmed that both early and peak RFD are more sensitive to neuromuscular fatigue and muscle damage than PF, making them more effective markers for monitoring acute changes in muscle function [12,30]. Smith et al. [31] examined explosive motions while standing up across ages and found that early RFD correlated with sit-to-stand power performance, but late RFD did not. Relation to power performance was also found in elite weightlifters by Zaras et al. [32], when leg press significantly increased RFD after training while PF remained same (*p* < 0.05), but their study evaluated a later force–time window, from 0 to 200 ms. As mentioned above, we can conclude that early RFD and overall RFD seem to have higher sensitivity and a stronger relationship with explosive performance, muscle fatigue, and damage than PF, as these indicators reflect the ability to generate force quickly, which is significant in rapid movements [12,29,30,31,32].

This low association between early RFD and PF emphasizes the utility of RFD-specific testing protocols and permits comparison of key throwing performance indicators with different values of athletes’ maximal strength [33,34]. Additionally, early RFD seems to respond differently to training than PF. High-load resistance training showed significant increases (9.2–14.6%; d = 2.71–4.16; *p* ≤ 0.001) in explosive/early force production, while moderate-load resistance training mainly increases PF (7.7 ± 11.8%, d = 2.02, *p* = 0.003) but not RFD (0.2–2.7%; d = 0.00–0.88; *p* > 0.05) [35]. Similar findings were reported by Zushi et al. [29], who showed early RFD to be the best parameter of improved (*p* = 0.03, d = 0.84) vertical jump through explosive force production in comparison to late RFD, average RFD, or PF.

The lack of significant differences between ASH Y and T can be explained by the deliberate trunk instability created in both positions within the test protocol (hand behind back unable to stabilize on the support surface as seen in Figure 1). Another explanation is in accordance with the previous findings in position-dependent relationships in muscle synergies within electromyographical (EMG) analysis, as different shoulder abduction angles activate muscles and their muscle synergies differently [36,37]. The synergy between anterior and posterior muscle groups shifts with abduction angle, affecting both force generation and joint stability during flexion and abduction mainly between the pectoralis major, anterior/posterior deltoid, latissimus dorsi, trapezius, and rotator cuff muscles [22,36,37]. These muscle synergies are yet unknown within ASH test positions; however, it is assumed that different muscle synergies were potentially responsible for generating similar forces despite no differences in ASH Y and T in strength performance in baseball athletes and should be examined in future in larger groups and different overhead sports. However, when individual relationships and muscle synergies between each of ASH test positions are confirmed, focused strength training within particular directions and planes of movement may enhance specific strength performance in each selected ASH test position. However, the fact that ASH T values were relatively high (not different to ASH Y) in baseball pitchers, corresponds with the idea that the ASH T position is the closest to ball release angle due to trunk side bend and 90° to 100° shoulder abduction [5]. Future research should examine ASH I’s correspondence to baseball pitching in terms of rate, magnitude, and timing.

This study confirmed and expanded previous research which established the excellent reliability of the ASH test for PF (ICC = 0.94–0.98) but not did not report early RFD results [17]. Excellent results in PF (ICC = 0.86–0.93) were also found in this study, supporting its use in maximal strength profiling [38]. However, good to moderate results (ICC = 0.77–0.81) in early RFD suggested higher intra-individual variability. While higher ICC results and moderate to strong correlations (ICC = 0.82–0.97; r = 0.62–0.89) between PF and RFDs (50, 100, 150, 200, and 250 ms) were found in isometric mid-thigh pull by Comfort et al. [38], our ASH test results confirmed that early RFD seems to be more sensitive to individual test protocol, effort quality of neuromuscular activation, technical precision, and execution consistency compared to PF results [35,39]. Thus, it is vital to explain and motivate the athlete to execute each repetition as fast as possible and in consistent maximal effort when evaluating early RFD.

Since pitch count limits are based on general recommendations rather than specific athlete data, ongoing monitoring of individual athlete recovery and performance is vital for effective injury prevention and workload management. The high volume of pitching in high-school baseball (>400) was related to in-season weakness of supraspinatus and lowered strength progression from one year to the next [40], while another study of collegiate pitchers found an average of 7735 throws per player for the entire season and showed moderate correlation (r = 0.72; *p* = 0.004) between arm soreness and pitch volume [41]. Additionally, the upper limit of pitch volume recommended for 17- to 18-year-old athletes by the Major League Baseball (MLB) Pitch Smart guidelines is 105 pitches per day/game, and for ages 19 to 22, it is 120 pitches, but the number of pitches often exceeds 120, going up to 140 counts depending on individual leagues and nationalities [42,43]. Additionally, MLB guidelines recommend that pitch count should be adequately reduced according to the number of rest days after reaching a daily maximum (0-day rest 1–30; 1-day rest 31–45; 2-day rest 46–60; 3-day rest 61–80; 4-day rest 81–105; 5-day rest 106+), also recommending to “keep track of the pitching volumes throughout the day and the year”, “setting and follow[ing] pitch-count limits and required rest periods”, and “monitor[ing] for other signs of fatigue” [43]. However, despite its high popularity, that information is not supported by research, and other authors also emphasize the lack of more exact data of adult athlete pitching volume recommendations, workload considerations in relation to throwing performance, reduction in strength, and injury risks in elite baseball [44,45]. As mentioned, we may conclude that pitching volume varies from age, pitching level or overall training strategy, while monitoring of workload and changes in training adaptation are always recommended for early detection of abnormality and injury prevention.

From a strength deficit perspective, it seems there is no recognized cutoff threshold, nor is there a strict percentage decrease universally used to detect early shoulder abnormality. Reductions of 10 to 30% in PF or RFD [30,46], bilateral asymmetries from 10 to 15% [47,48], or unilateral strength ratios < 0.75% [49] are often considered clinically meaningful and may indicate increased injury risk. SWC, in this research study, reached about 4% in PF and PF/BM, while MDC was around 15%. Higher SWC and MDC (15% and 50% approx.) in the RFD parameter’s context confirmed the higher variability in individual performance in explosive strength [20]. Thus, RFD appears to be a more sensitive early marker than PF for detection of neuromuscular deficits [12,30]. However, isometric shoulder strength assessments in baseball often require minimal equipment, are fast to perform, and can be used both in the field and laboratory for screening injury risk, monitoring training effects, and identifying strength deficits or imbalances [17,50,51,52]. It is useful for preseason assessments and ongoing athlete monitoring, though its direct link to throwing performance may vary by age and test type [51].

The variability within RFD on maximal isometric testing may also be indicative of the lack of attention to specific training of upper-limb RFD away from sports-specific actions. In our experience of serial testing of an athlete’s explosive capability in an ASH test, there is relatively natural progression and improvement from repeated exposures in its early introduction. This not only makes familiarization important prior to ASH test use as an in-season monitoring tool but also aligns with lower-body research demonstrating that high maximal squat does not infer explosive jump capability in less trained individuals [35,39]. PF and explosive force are distinct from one another. ASH test RFD stability in a single session may be an indicator of an athlete’s ability to repeat explosive actions, likely a product of prior training exposure.

The importance of using international-level throwers meant that access to a relatively lower number of subjects likely narrowed the variability of performance. Although all subjects were familiarized, it was not possible to control the 72 h prior to testing, which could have influenced ability to produce explosive force. We performed all three tests in a consistent order, with ASH T last, which may have affected RFD outputs. Moreover, this study did not evaluate relationships between throwing performance and ball velocity or explore the effect of injury history. While participants were excluded if they presented with current pain or musculoskeletal injury, retrospective injury histories were not collected in this study. Therefore, findings cannot be directly associated with prior injuries or adaptations resulting from them. Future research should examine whether specific positional ASH measures differ between previously injured/uninjured throwing athletes. A priori power analysis was not conducted; however, the study included all available national team baseball players. A post hoc power analysis was deliberately not performed, as methodological literature indicates that such analyses do not provide meaningful information about study adequacy or true statistical power [53,54]. Future studies are recommended to perform power analysis during study design. Additionally, our findings are limited to elite male baseball players; the absence of a control group or comparisons across competition levels limits its generalizability. Implementation of this in future research may enhance the relevance of early RFD. Analysis of female athletes, as well as more overhead sports-specific analysis, is recommended. The small sample size did not allow for separate analysis of pitchers vs. infield/outfield players, who may have different shoulder strength characteristics. Future studies with larger numbers of participants should examine positional differences to enhance the specificity of normative values and training recommendations. The results of various ASH test positions may be influenced by adaptation to specific arm positions within throwing, so analysis of ASH test position-based differences is also recommended in different international-level throwers. Additionally, anthropometric characteristics like chest circumference, limb length, and levelling of shoulder and elbow joints relative to the force plate were not systematically measured in this study. Although the ASH test protocol ensured consistent positioning across trials, these factors may influence intra-individual validity by altering lever arms and torque demands [22,25]. Future research should incorporate standardized anthropometric assessments and joint alignment measures to provide stronger evidence for the inter- and intra-individual consistency of ASH test outcomes. Exposure to consistent RFD-based training (e.g., jumping) is common in the lower body but less apparent in upper-body programs. Future research should examine different strength assessments and multi-joint movement differences in reliability of early RFD. The standard testing order of I, Y, T and completion of all three trials for a given position consecutively may have introduced potential order effects [17]. Future studies should examine whether changing the order of positions or interleaving trials between positions affect PF or early RFD. The window between 0 and 100 ms for early RFD evaluation was used to reflect the time demands of throwing, while total trial time was 3 s, but further research evidence explicitly related to rapid strength application capability in baseball is required. Future studies should examine whether shorter total trial durations or dynamic protocols yield comparable ecological validity.

## 5. Conclusions

This study found that the ASH I position (180° shoulder abduction) produced the highest results in PF and early RFD in elite baseball players, likely due to greater deltoid and rotator cuff contribution at higher abduction angles, which mirror the demands of the throwing motion. In contrast, the ASH Y (135°) and ASH T (90°) positions demonstrated similar outputs, suggesting that different shoulder angles recruit distinct muscle synergies but may not differ in maximal isometric strength for baseball population. The excellent reliability of PF supports its use for routine strength profiling, while the greater variability of early RFD highlights the need for repeated measures and careful interpretation. A decreasing force trend may signal cumulative fatigue throughout the season, reflecting maladaptation to high individual volumes of maximal throws. Practical application of these findings provides reference values for position-specific strength thresholds and highlights the importance of including multiple abduction angles in shoulder strength assessment. Regular assessment is an important, safe, time-efficient and relatively non-expensive method to detect early neuromuscular abnormalities, strength deficits, monitor training adaptations, and guide rehabilitation progressions in throwing athletes.

## Figures and Tables

**Figure 1 sports-13-00300-f001:**
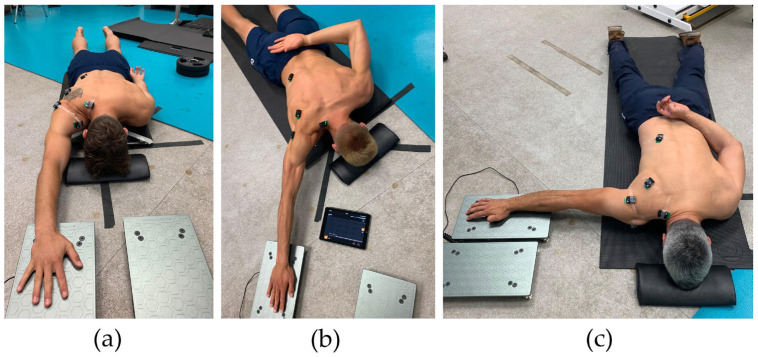
Isometric strength assessment by Athletic Shoulder (ASH) test using force plate system ForceDecks (Vald, Brisbane, Australia). Shoulder in three positions: (**a**) ASH I (180° shoulder abduction); (**b**) ASH Y (135° shoulder abduction); and (**c**) ASH T (90° shoulder abduction).

**Figure 2 sports-13-00300-f002:**
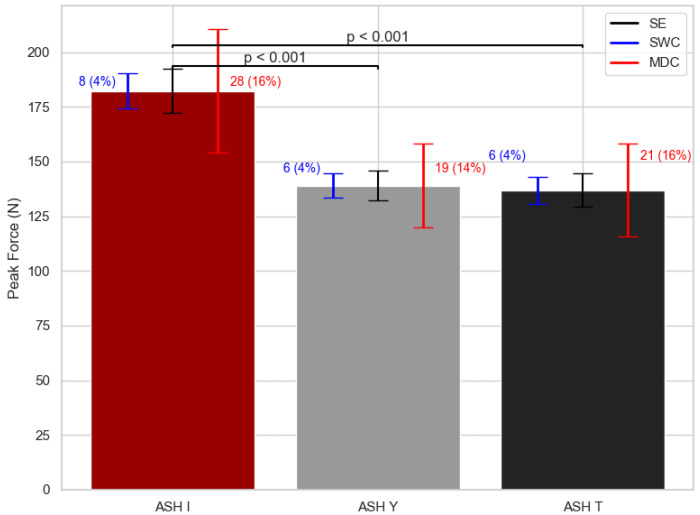
Peak force mean values with standard error (SE), smallest worthwhile change (SWC), and minimal detectable change (MDC) bars for each ASH test shoulder position. ASH I: Athletic Shoulder test in I position (180° shoulder abduction); ASH Y: Athletic Shoulder test in Y position (135° shoulder abduction); ASH T: Athletic Shoulder test in T position (90° shoulder abduction). Significant pairwise differences are marked with lines and *p*-values.

**Figure 3 sports-13-00300-f003:**
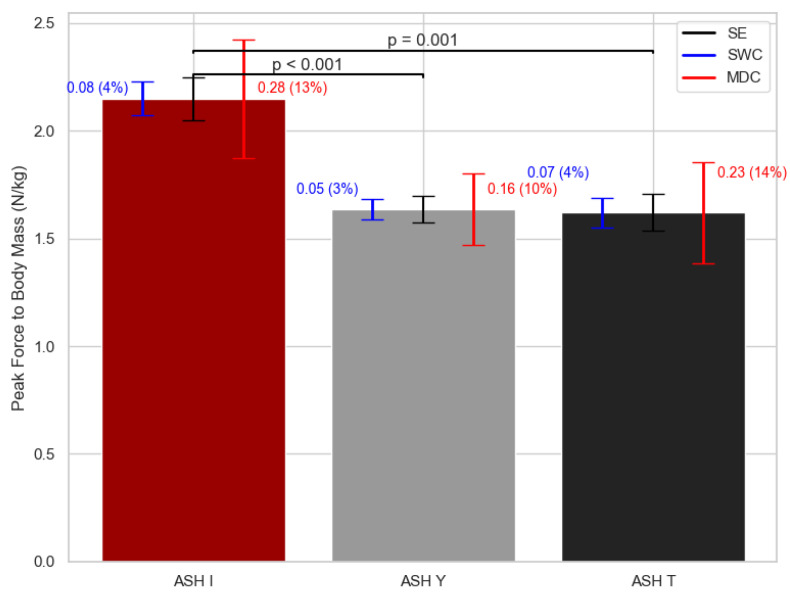
Peak force normalized to body mass mean values with standard error (SE), smallest worthwhile change (SWC), and minimal detectable change (MDC) bars for each ASH test shoulder position. ASH I: Athletic Shoulder test in I position (180° shoulder abduction); ASH Y: Athletic Shoulder test in Y position (135° shoulder abduction); ASH T: Athletic Shoulder test in T position (90° shoulder abduction). Significant pairwise differences are marked with lines and *p*-values.

**Figure 4 sports-13-00300-f004:**
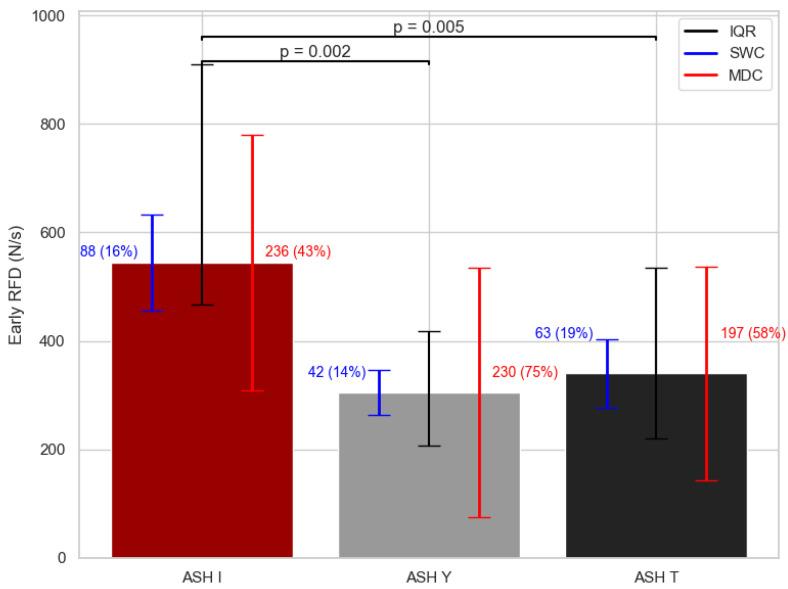
Early rate of force development (RFD) median values with interquartile range (IQR; 25th–75th), smallest worthwhile change (SWC), and minimal detectable change (MDC) error bars for each ASH test shoulder position. ASH I: Athletic Shoulder test in I position (180° shoulder abduction); ASH Y: Athletic Shoulder test in Y position (135° shoulder abduction); ASH T: Athletic Shoulder test in T position (90° shoulder abduction). Significant pairwise differences are marked with lines and *p*-values.

**Table 1 sports-13-00300-t001:** Descriptive peak force and early rate of force development results data of analysed baseball athletes in three shoulder ASH test positions.

Test	PF (N)	PF/BM (N/kg)	Early RFD (N/s)
	Mean (SD)	Mean (SD)	Median (IQR 25th–75th)
ASH I	182 (41)	2.15 (0.39)	545 (468–910)
ASH Y	139 (28)	1.64 (0.28)	305 (208–418)
ASH T	137 (31)	1.62 (0.34)	340 (220–535)

PF: peak force; BM: body mass; N: newtons; kg: kilograms; s: seconds; RFD: rate of force development; IQR: interquartile range; ASH I: Athletic Shoulder test in I position; ASH Y: Athletic Shoulder test in Y position; ASH T: Athletic Shoulder test in T position.

## Data Availability

The raw data supporting the conclusions of this article will be made available by the authors on request.

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
