# Peer review of "Early Rate of Force Development and Maximal Strength at Different Positions of the Athletic Shoulder Test in Baseball Players"

_sports, 2025, doi:10.3390/sports13090300_

Round 1

Reviewer 1 Report

Comments and Suggestions for Authors

Thank you for the opportunity to evaluate this scientific work. In an effort to synthesize, I recommend the following aspects to the authors for correction:

Introduction
The text mentions a lack of previous studies in baseball, but does not provide sufficient justification for the comparative relevance between the three positions tested.
Materials and Methods
The absence of a control group or a comparison between athletes of different levels I believe limits the ability to generalize the results.
Results
The high variation of the RFD is not clearly indicated and how this could influence the practical validity of the test.
Discussion, Conclusions, Bibliography
No comments

Author Response

Author's Reply to the Review Report (Reviewer 1)

Thank you for the opportunity to evaluate this scientific work. In an effort to synthesize, I recommend the following aspects to the authors for correction:

Introduction
The text mentions a lack of previous studies in baseball, but does not provide sufficient justification for the comparative relevance between the three positions tested.

Author response: Thank you for this important insight. According to second reviewer and your recommendation about connecting the ecological validity of ASH test position, biomechanics, and comparative relevance between three positions tested. We have updated the Introduction part with new text as follows: “In baseball pitching, shoulder abduction angles typically range from approximately 90° at ball release to 170 or 180° during maximal shoulder external rotation in the late cocking phase [5, 6]. At such a higher angle, neuromuscular coordination between deltoid muscles and rotator cuff muscles is important for maintaining glenohumeral stability and effective force transfer [21]. Each of the ASH test positions corresponds to a portion of this range, as ASH I (180° of shoulder abduction) reflects the maximal abduction seen in late cocking, ASH Y (135° of shoulder abduction) reflects the transition to acceleration, and ASH T (90° of shoulder abduction) reflects the arm position at ball release [5, 22]. This alignment suggests that the ASH test may offer greater ecological validity for baseball than short lever internal and external shoulder rotation assessment, as it replicates the long lever and high shoulder abduction during pitching. While previous research in volleyball players has linked ASH I early RFD to serve velocity [18], baseball differs in that the ball must be re-leased after being carried through an optimal acceleration trajectory, potentially creating distinct neuromuscular demands that justify specific positional strength assessment. Comparing three ASH test positions in baseball athletes can identify angle specific results or deficits, inform targeted conditioning, and improve the sensitivity of monitoring programs for performance optimization and injury prevention.”

Materials and Methods
The absence of a control group or a comparison between athletes of different levels I believe limits the ability to generalize the results.

Author response: Thank you for this notification, while we have not used control group, we added this to study limitation: „Additionally, our findings are limited to elite male baseball players; absence of a control group or comparisons across competition levels limits generalizability.”

Results
The high variation of the RFD is not clearly indicated and how this could influence the practical validity of the test.

Author response: Thank you for this comment, we have added connection to this in new conclusion paragraph as follows:” …The excellent reliability of PF supports its use for routine strength profiling, while the greater variability of early RFD highlights the need for repeated measures and careful interpretation. …”

Discussion, Conclusions, Bibliography
No comments

Author response: Thank you

Reviewer 2 Report

Comments and Suggestions for Authors

sports-3750100

Reviewer comments

In the submitted manuscript, the authors examined force output parameters in baseball players when tested with the Athletic Shoulder test in 3 different positions of the shoulder using uniaxial force-plates. Results revealed that peak force, relative to body mass peak force and maximum RFD at the first 100ms of the test was significantly higher when the shoulder was in 180 deg abduction, and that peak force had excellent reliability whereas RFD was moderate-to-good. It was concluded that ASH should be conducted in different shoulder positions.

The submission is well within the scope of the Journal. However, there are several topics that need to be addressed, as mentioned in the following General and Specific Comments.

General Comments

  • Introduction: Further elaboration is required relating to the importance of testing the different shoulder abduction positions tested in this study in relation to baseball throwing technique. Also, given the example of previous studies utilizing ASH in volleyball players, when the task is to have an impact with the ball, in baseball you have to release the ball after carrying it with an optimal trajectory of acceleration. The rationale for selecting ASH for testing the force output capabilities of throwing overhead athletes and baseball players in particular under the perspective of the ecological validity of the test should be further introduced.
  • Methods: Developing further the previous comment, why the 0-100ms RFD was used as testing parameter in a 3 s test? [note: this information is derived from the abstract and the Introduction, as this information lacks from the “2.2 Procedures” subsection]. It would be of interest to get the rationale for the selected duration of the test compared to a duration of i.e., 1 s and its ecological validity against the ball release action in ball throwing and ball pitching. Note also that the Numbering of the subsections are inconsistent (2.2, followed by 2.3.1, and then is 2.3). In general, more information should be provided for data acquisition (i.e., sampling frequency) and the data analysis (filtering of the force signals, selection of the onset of the trial), with the latter being essential as the hypothesis of the study is tested based on the data analysis procedure.
  • Discussion/Limitations: it is stated that an exclusion criterion was the absence of injuries, but in L228 there is an argument that the findings were not related to injury history. Could you please clarify that? Also, it is recommended to elaborate on the ecological validity of the measures relating to arm mechanics in baseball throwing/pitching.
  • Conclusions: although practical implications for coaches are provided, the link between the concluding statements in both the abstract and the text about the return-to-play decision and seasonal variability and the measures undergone in the present study is weak. It is recommended to focus on the main outcomes that are derived from the present findings.

Specific Comments

Introduction

  • L69: it is recommended not to split the paragraph here. Nevertheless, split the paragraph when the purpose of the study is stated.

Materials and Methods

  • L80: Provide details regarding the IRB approval.
  • L83: Provide the statistical power based on the recruited participants.
  • L83: Comment whether the positional differences (pitchers vs. fielders/basemen) could tamper the results of the study.
  • L83: define if the “position players” were infield or outfield players.
  • LL105-106: what was the height difference between the center of the elbow joint and the surface of the force-plate? Was it adjusted based on anthropometrics (i.e., chest circumference)?
  • LL110-111: Again, information is missing. In the abstract (L20) it is mentioned that the dominant arm was measured. There is no such information here. Nevertheless, how was arm dominancy considered/evaluated?
  • L111: Provide the rationale for not providing the I, Y, and T ASH tests

Discussion

  • L232: international level throwers.

Conclusions

  • This section is replicating the results of the study rather than highlighting the mechanism responsible for the findings of the study, and the main outcome of the study. It is suggested to revise this part of the manuscript, emphasizing also on the practical implications and recommendations for practitioners and coaches.

Author Response

Author's Reply to the Review Report (Reviewer 2)

In the submitted manuscript, the authors examined force output parameters in baseball players when tested with the Athletic Shoulder test in 3 different positions of the shoulder using uniaxial force-plates. Results revealed that peak force, relative to body mass peak force and maximum RFD at the first 100ms of the test was significantly higher when the shoulder was in 180 deg abduction, and that peak force had excellent reliability whereas RFD was moderate-to-good. It was concluded that ASH should be conducted in different shoulder positions.

The submission is well within the scope of the Journal. However, there are several topics that need to be addressed, as mentioned in the following General and Specific Comments. 

General Comments

  • Introduction: Further elaboration is required relating to the importance of testing the different shoulder abduction positions tested in this study in relation to baseball throwing technique. Also, given the example of previous studies utilizing ASH in volleyball players, when the task is to have an impact with the ball, in baseball you have to release the ball after carrying it with an optimal trajectory of acceleration. The rationale for selecting ASH for testing the force output capabilities of throwing overhead athletes and baseball players in particular under the perspective of the ecological validity of the test should be further introduced.
  • Author response: Thank you for such an important recommendation. We have updated the Introduction with new text as follows: “In baseball pitching, shoulder abduction angles typically range from approximately 90° at ball release to 170 or 180° during maximal shoulder external rotation in the late cocking phase [5, 6]. At such a higher angle, neuromuscular coordination between deltoid muscles and rotator cuff muscles is important for maintaining glenohumeral stability and effective force transfer [21]. Each of the ASH test positions corresponds to a portion of this range, as ASH I (180° of shoulder abduction) reflects the maximal abduction seen in late cocking, ASH Y (135° of shoulder abduction) reflects the transition to acceleration, and ASH T (90° of shoulder abduction) reflects the arm position at ball release [5, 22]. This alignment suggests that the ASH test may offer greater ecological validity for baseball than short lever internal and external shoulder rotation assessment, as it replicates the long lever and high shoulder abduction during pitching. While previous research in volleyball players has linked ASH I early RFD to serve velocity [18], baseball differs in that the ball must be re-leased after being carried through an optimal acceleration trajectory, potentially creating distinct neuromuscular demands that justify specific position strength assessment. Comparing three ASH test positions in baseball athletes can reveal specific position optimal results or deficits, inform targeted conditioning, and improve the sensitivity of monitoring programs for performance optimization and injury prevention.”

  • Methods: Developing further the previous comment, why the 0-100ms RFD was used as testing parameter in a 3 s test? [note: this information is derived from the abstract and the Introduction, as this information lacks from the “2.2 Procedures” subsection]. It would be of interest to get the rationale for the selected duration of the test compared to a duration of i.e., 1 s and its ecological validity against the ball release action in ball throwing and ball pitching. Note also that the Numbering of the subsections are inconsistent (2.2, followed by 2.3.1, and then is 2.3). In general, more information should be provided for data acquisition (i.e., sampling frequency) and the data analysis (filtering of the force signals, selection of the onset of the trial), with the latter being essential as the hypothesis of the study is tested based on the data analysis procedure.
  • Author response: Thank you very much and we appreciate the request for clarifications. We selected 0-100 ms as it represents the early explosive force production phase, critical in throwing where arm acc occurs within 150 ms. Your point about 1 s and stronger relevance to ball release action in ball throwing is important and true. However, standard MVC durations and ASH test protocol used by research and clinicians use the 3 s force output, to ensure max peak force was also achieved within the same effort. Early phase examines the portion of the force curve most relevant to rapid action demand, while peak force and maximum participants force capacity would lay overall maximum strength profile, which is shown to be also closely related to performance and injury prevention, fatigue resistance. Use of PF was also confirmed by its reliability vs. reliability of early RFD. Regarding ecological validity, the high abduction, long lever isometric performance in ASH test should mirror the shoulder joint loading of the throwing, making early RFD a relevant indicator despite the static nature of the test. We have corrected subsection numbering for clarity and added details on data acquisition, signal processing, and onset detection in the revised Methods section as follows:
  • Author response: We have corrected the numbering of the “2.2.1 Isometric shoulder flexion strength assessment”
  • Author response: We have corrected the Procedures and added this:”Each maximal voluntary contraction was held for approximately 3 s to allow assessment of both RFD (0–100 ms) and PF within the same trial. The 0–100 ms window was selected because it reflects the early explosive force production phase, which is most relevant to the rapid arm acceleration in baseball pitching and throwing, typically occurring within 150 ms [20].”
  • Author response: We have added the new sub section 2.4 Data Processing as follows:” Force-time data were collected at a sampling frequency of 1000 Hz (ForceDecks, Vald, Australia) and exported for analysis. Signals were filtered using a 4th order Butterworth low-pass filter with a 20 Hz cutoff. Contraction onset was defined as the point where force exceeded baseline by more than 5 N within a 0.2-second window. Early RFD was calculated as the change in force from contraction onset to the force value at 100 ms, divided by the corresponding time interval. PF was defined as the highest instantaneous force value recorded within the trial.”
  • Author response: We added this according to ecological validity and shorter trial times for future recommendations in study limits at the end of the Discussion as follows: „The window between 0 to 100 ms for early RFD evaluation was used to reflect the time demands of throwing, while total trial time was 3 s. Future studies should examine whether shorter total trial durations or dynamic protocols yield comparable ecological validity.”

  • Discussion/Limitations: it is stated that an exclusion criterion was the absence of injuries, but in L228 there is an argument that the findings were not related to injury history. Could you please clarify that? Also, it is recommended to elaborate on the ecological validity of the measures relating to arm mechanics in baseball throwing/pitching.
  • Author response: Thank you for this point. The exclusion criteria ensured that no participants had current pain or musculoskeletal injuries at the time of testing. However, we did not collect detailed retrospective injury history, so our results cannot be directly related to past injuries. We have clarified this in study limits. We assume that the argument was related to the L328 (not 228). We have written a new sentence as follows: „While participants were excluded if they presented with current pain or musculoskeletal injury, retrospective injury histories were not collected in this study. Therefore, findings cannot be directly associated with prior injuries or adaptations resulting from them. Future research should examine whether specific position ASH measures differ between previously injured/uninjured throwing athletes.”
  • Author response: According to ecological validity, we have added this to the beginning of the discussion as follows: “The ASH test’s high abduction and long lever arm positions replicate the shoulder joint loading pattern and muscle coordination demands observed in the late cocking and acceleration phases of baseball pitching, where shoulder abduction angles range from approximately 90° to 180° [5, 22, 23]. These shoulder positions need similar contributions from the deltoid and rotator cuff muscles and scapular stabilizers [24, 25], supporting the ecological validity of ASH test resulting PF and early RFD measures for performance monitoring, rehabilitation, and return-to-play decision making in baseball.”
  • We have also added the new and one of the most important, from our clinical perspective of working with real athletes, closing thoughts, which blend the outcomes and authors insights and new thoughts about practical finding and understanding the results for practitioners, who work with thrower athletes: „ The variability within RFD on maximal isometric testing may also be indicative of the lack of attention to specific training of upper limb RFD away from sports specific actions. In our experience of serial testing of an athlete’s explosive capability in an ASH test, there is relatively natural progression and improvement from repeated exposures in its early introduction. This not only makes familiarisation important prior to ASH test use as an in-season monitoring tool but also aligns with lower body research that demonstrates high maximal squat does not infer explosive jump capability in less trained individuals [35, 39]. PF and explosive force are distinct from one another. ASH test RFD stability in a single session may be an indicator of an athlete’s ability to repeat explosive actions, likely a product of prior training exposure.”

  • Conclusions: although practical implications for coaches are provided, the link between the concluding statements in both the abstract and the text about the return-to-play decision and seasonal variability and the measures undergone in the present study is weak. It is recommended to focus on the main outcomes that are derived from the present findings.
  • Author response: Thank you, we have written new conclusion as stated later in the specific comments. While also updating the abstract conclusions and omitted the return to play and seasonal variability assumptions as follows: “Maximal isometric shoulder strength and explosiveness were highest at 180° abduction in baseball athletes, with no significant difference between 135° and 90°. PF demonstrated excellent reliability, while early RFD showed moderate to good reliability and higher variability, highlighting the need for repeated measures. These findings provide specific position reference values and support the inclusion of multiple abduction angles in shoulder strength assessment to detect neuromuscular deficits and monitor training adaptations in baseball athletes.”

 Specific Comments

Introduction

  • L69: it is recommended not to split the paragraph here. Nevertheless, split the paragraph when the purpose of the study is stated.
  • Author response: Thank you for this enhancement, we have split the paragraph higher in the text, starting with the “Thus, this study examined…”

 Materials and Methods

  • L80: Provide details regarding the IRB approval.
  • Author response: Thank you for notification. The information about IRB was added at the end of the Participants paragraph as follows: “The study was conducted in accordance with the Declaration of Helsinki and approved by the Institutional Ethical Committee of the Liverpool Hope University under n. S11-06-19PA049. Informed consent was obtained from all subjects involved in the study.”

  • L83: Provide the statistical power based on the recruited participants.
  • Author response: Thank you for mentioning the power analysis, power analysis was not performed prior to the data collection, as absolute available participants of national baseball team players were recruited. After consultation with our statistician, we have been recommended to not perform post-hoc power analysis as this analysis doesn't provide any information about the power of the study or the adequacy of the sample size. For example, as Zhang et al. Post hoc power analysis: is it an informative and meaningful analysis? or Heckman et al. Post Hoc Power Calculations: An Inappropriate Method for Interpreting the Findings of a Research Study. The information about the power analysis omission and future recommendation was added to the study limitation in the end of the discussion part as follows: “A priori power analysis was not conducted; however, the study included all available national team baseball players. Future studies are recommended to perform power analysis during study “

  • L83: Comment whether the positional differences (pitchers vs. fielders/basemen) could tamper the results of the study.
  • Author response: Thank you for this important comment. We have commented this in study limitations at the end of the discussion as follows:” The small sample size did not allow for separate analysis of pitchers vs. fielders/basemen, which may have different shoulder strength characteristics. Future studies with larger number of participants should examine positional differences to enhance the specificity of normative values and training recommendations.”

  • L83: define if the “position players” were infield or outfield players.
  • Author response: Thank you for detailed insight into problematics. While we understand the reviewer’s detailed appreciation of subtleties of infield and outfield players, we have chosen to make a distinction within this paper of pitchers and position players. Future studies may further explore this distinction, and it was mentioned within the study limitation as follows: “The small sample size did not allow for separate analysis of pitchers vs. fielders/basemen, which may have different shoulder strength characteristics. Future studies with larger number of participants should examine positional differences to enhance the specificity of normative values and training recommendations.”

  • LL105-106: what was the height difference between the center of the elbow joint and the surface of the force-plate? Was it adjusted based on anthropometrics (i.e., chest circumference)?
  • Author response: Thank you another detail insight. We appreciate the level of detail here relating to the effect of individual anthropometrics on the test, however as a field based test the player’s test / retest values mean that intra-player chest height differences will be more likely to be consistent. Also, we are used to comparing force plate scores for other tests without this level of detail (e.g., CMJ, Isometric hamstring 90/90). Further work will normalise for lever length, and we already look at body weight normalised scores (N/kg and N/kg/s). We understand and appreciate the thought process but feel that the efficiency of data collection in a field based test is a priority for us.

  • LL110-111: Again, information is missing. In the abstract (L20) it is mentioned that the dominant arm was measured. There is no such information here. Nevertheless, how was arm dominancy considered/evaluated?
  • Author response: Thank you for mentioning dominant arm selection. We have added this in the Procedures paragraph as follows:” Only the dominant upper limb was tested, defined as the athlete’s preferred throwing arm, which was identified by verbal questioning.

  • L111: Provide the rationale for not providing the I, Y, and T ASH tests
  • Author response: Thank you for this comment. We have put the rationale behind the use of the I, Y and T order, and added new recommendation about the order effect in the study limits as follows:” The order of testing positions was consistently I, Y, T, as recommended by the original ASH test protocol [17] and commonly applied by clinicians.” And “The standard testing order of I, Y, T and completion of all three trials for a given position consecutively may have introduced potential order effects (17). Future studies should ex-amine whether changing the order of positions or interleaving trials between positions affect PF or early RFD.”

Discussion

  • L232: international level throwers.
  • Author response: Thank you for enhancing understanding of this sentence and clarifying this. We believe you were relating this to the L323 and not 232, or if yes, we apologize. We have changed the words as follows to original L323: “Results of various ASH test positions may be influenced by adaptation to specific arm positions within throwing, so analysis of ASH test positional differences is also recommended in different international level throwers.”

Conclusions

  • This section is replicating the results of the study rather than highlighting the mechanism responsible for the findings of the study, and the main outcome of the study. It is suggested to revise this part of the manuscript, emphasizing also on the practical implications and recommendations for practitioners and coaches.
  • Author response: Thank you this important comment. We have written new conclusion paragraph:” This study found that the ASH I position (180° shoulder abduction) produced the highest results in PF and early RFD in elite baseball players, likely due to greater deltoid and rotator cuff contribution at higher abduction angles, which mirror the demands of the throwing motion. In contrast, the ASH Y (135°) and ASH T (90°) positions demonstrated similar outputs, suggesting that different shoulder angles recruit distinct muscle synergies but may not differ in maximal isometric strength for baseball population. The excellent reliability of PF supports its use for routine strength profiling, while the greater variability of early RFD highlights the need for repeated measures and careful interpretation. A decreasing force trend may signal cumulative fatigue throughout the season reflecting maladaptation to high individual volumes of maximal throws. Practical application of these findings provides reference values for position-specific strength thresholds and highlight the importance of including multiple abduction angles in shoulder strength assessment. Regular assessment is an important, safe, time-efficient and relatively non-expensive method to detect early neuromuscular abnormalities, strength deficits, monitor training adaptations, and guide rehabilitation progressions in throwing athletes.”

Round 2

Reviewer 1 Report

Comments and Suggestions for Authors

The authors have made the necessary corrections, and the version indicated by them is appropriate. 

Author Response

Dear reviewer, 

thak you very much for your work and comments in the first round. We are glad to hear you are satisfied.

We wish all the best in the future. 

Reviewer 2 Report

Comments and Suggestions for Authors

sports-3750100-R1

Reviewer comments

In the resubmitted version of the manuscript, the authors did an exceptional job to address the comments raised in the initial round of reviewing. Nevertheless, some topics still require a further address by the authors.

Comments

  1. Title: It is suggested, since only baseball players were examined, to modify as: “Early Rate of Force Development and Maximal Strength at Different Positions of the Athletic Shoulder Test in Baseball Players”.
  2. Paragraph 2 spans from L55 to L86; consider splitting.
  3. L103: As the throwing demands differ significantly between infield and outfield players, it is recommended mentioning the number of the infield/basemen and outfield players that participated in the study.
  4. L129: The anthropometrics and the leveling of the upper arm joints (shoulder+elbow) with the surface of the forceplate are essential for the intra-individual validity of the measures as torque is minimized. It is proposed to provide and report further evidence about the consistency of the anthropometrics in both inter-test and intra-individual level.
  5. LL135-139, Choice of the 0-100 ms window for RFD: further research evidence explicitly related to rapid strength application capability in baseball is required.
  6. L376: “international level throwers” instead of “elite throwers”.
  7. L387: Add the rationale and the respective references as in your response to the initial Reviewer’s report (comment for L83 of the initial submission).

Author Response

Dear reviewer, let us thank you very musch for deep insight and great help within exploration the possible updates and such an enhacements to our manuscript. You helped us to elevate the research quality and its interpretation to the readers and scientific sphere. We believe we answered all your questions in full scale, while lot of them was added in study limitations. Thank you again. 

Responses to the reviewers comments:

In the resubmitted version of the manuscript, the authors did an exceptional job to address the comments raised in the initial round of reviewing. Nevertheless, some topics still require a further address by the authors.

Comments

  1. Title: It is suggested, since only baseball players were examined, to modify as: “Early Rate of Force Development and Maximal Strength at Different Positions of the Athletic Shoulder Test in Baseball Players”.

Authors response: Thank you for interesting update. We have changed the ending of the title.

  1. Paragraph 2 spans from L55 to L86; consider splitting.

Authors response: Thank you for the enhancement, we have made the split at L70.

  1. L103: As the throwing demands differ significantly between infield and outfield players, it is recommended mentioning the number of the infield/basemen and outfield players that participated in the study.

Authors response: Thank you for this update. We have retrospectively updated the numbers of players in study Methods section, while keeping the study limitation mentioning about further exploration of differences between player positions.

  1. L129: The anthropometrics and the leveling of the upper arm joints (shoulder+elbow) with the surface of the forceplate are essential for the intra-individual validity of the measures as torque is minimized. It is proposed to provide and report further evidence about the consistency of the anthropometrics in both inter-test and intra-individual level.

Authors response: Thank you for this insight. We will manage to incorporate this problematics and further study design in further studies, as we are focusing on exploring the ASH test widely. We also added this to the future study recommendations as follows: „Additionally, anthropometric characteristics like chest circumference, limb length and levelling of shoulder and elbow joints relative to the forceplate were not systematically measured in this study. Although the ASH test protocol ensured consistent positioning across trials, these factors may influence intra-individual validity by altering lever arms and torque demands [22,25]. Future research should incorporate standardized anthropo-metric assessments and joint alignment measures to provide stronger evidence for the inter- and intra-individual consistency of ASH test outcomes.”

  1. LL135-139, Choice of the 0-100 ms window for RFD: further research evidence explicitly related to rapid strength application capability in baseball is required.

Authors response: Thank you, this was added to the end of study limitations within the mentioning of the 100 ms window.

  1. L376: “international level throwers” instead of “elite throwers”.

Authors response: Thank you, we updated this sentence.

  1. L387: Add the rationale and the respective references as in your response to the initial Reviewer’s report (comment for L83 of the initial submission).

Authors response: Thank you for this clarification, we have updated this limit as follows: “A priori power analysis was not conducted; however, the study included all available national team baseball players. A post-hoc power analysis was deliberately not performed, as methodological literature indicates that such analyses do not provide meaningful information about study adequacy or true statistical power (Zhang et al., 2019; Heckman et al., 2022). Future studies are recommended to perform an a priori power analysis during study design to guide sample size estimation.”
